# High Fructose Intake Contributes to Elevated Diastolic Blood Pressure in Adolescent Girls: Results from The HELENA Study

**DOI:** 10.3390/nu13103608

**Published:** 2021-10-15

**Authors:** Laurent Béghin, Inge Huybrechts, Elodie Drumez, Mathilde Kersting, Ryan W Walker, Anthony Kafatos, Denes Molnar, Yannis Manios, Luis A Moreno, Stefaan De Henauw, Frédéric Gottrand

**Affiliations:** 1Univ. Lille, Inserm, CHU Lille, U1286—INFINITE and CIC-1403, F-59000 Lille, France; Frederic.gottrand@chu-lille.fr; 2Department of Public Health and Primary Care, Faculty of Medicine and Health Sciences, Ghent University, B-9000 Ghent, Belgium; HuybrechtsI@iarc.fr (I.H.); Stefaan.DeHenauw@ugent.be (S.D.H.); 3Dietary Exposure Assessment Group, International Agency for Research on Cancer, F-69000 Lyon, France; 4Univ. Lille, CHU Lille, ULR 2694—METRICS: Évaluation des Technologies de Santé et des Pratiques Médicales, F-59000 Lille, France; elodie.drumez@chu-lille.fr; 5CHU Lille, Department of Biostatistics, F-59000 Lille, France; 6Research Department of Child Nutrition, Pediatric University Clinic, Ruhr-University Bochum, D-44791 Bochum, Germany; mathilde.kersting@ruhr-uni-bochum.de; 7Department of Environmental Medicine and Public Health, Icahn School of Medicine at Mount Sinai, New York, NY 10029, USA; ryan.walker@mssm.edu; 8Preventive Medicine and Nutrition Clinic, University of Crete School of Medicine, G-14122 Crete, Greece; kafatos@med.uoc.gr; 9Department of Pediatrics, University of Pecs, H-7600 Pecs, Hungary; denes.molnar@aok.pte.hu; 10Department of Nutrition and Dietetics, University of Harakopio, G-10431 Athens, Greece; manios@hua.gr; 11GENUD (Growth, Exercise, Nutrition and Development) Research Group Escuela Universitaria de Ciencas de la Salud, Universidad de Zaragoza, S-50009 Zaragoza, Spain; lmoreno@unizar.es

**Keywords:** pure fructose consumption, adolescent, blood pressure

## Abstract

Background: The association between high fructose consumption and elevated blood pressure continues to be controversial, especially in adolescence. The aim of this study was to assess the association between fructose consumption and elevated blood pressure in an European adolescent population. Methods: A total of 1733 adolescents (mean ± SD age: 14.7 ± 1.2; percentage of girls: 52.8%) were analysed from the Healthy Lifestyle in Europe by Nutrition in Adolescence (HELENA) study in eight European countries. Blood pressure was measured using validated devices and methods for measuring systolic blood pressure (SBP) and diastolic blood pressure (DBP). Dietary data were recorded via repeated 24 h recalls (using specifically developed HELENA–DIAT software) and converted into pure fructose (monosaccharide form) and total fructose exposure (pure fructose + fructose from sucrose) intake using a specific fructose composition database. Food categories were separated at posteriori in natural vs. were non-natural foods. Elevated BP was defined according to the 90th percentile cut-off values and was compared according to tertiles of fructose intake using univariable and multivariable mixed logistic regression models taking into account confounding factors: centre, sex, age and z-score–BMI, MVPA (Moderate to Vigorous Physical Activity) duration, tobacco consumption, salt intake and energy intake. Results: Pure fructose from non-natural foods was only associated with elevated DBP (DBP above the 10th percentile in the highest consuming girls (OR = 2.27 (1.17–4.40); *p* = 0.015) after adjustment for cofounding factors. Conclusions: Consuming high quantities of non-natural foods was associated with elevated DBP in adolescent girls, which was in part due to high fructose levels in these foods categories. The consumption of natural foods containing fructose, such as whole fruits, does not impact blood pressure and should continue to remain a healthy dietary habit.

## 1. Introduction

The introduction of industrial foods and new manufacturing food process at the beginning of 20th century has modified sources of fructose in humans. For centuries, the main source of fructose in humans was the consumption of natural foods containing fructose such as honey, fruits and vegetables intrinsically rich in the pure monosaccharide form (α-D-glucopyranose) [1]. Since the substitution of sucrose for fructose in many foods and beverages, the consumption of extrinsic fructose has increased in parallel with increasing industrially manufactured or confectionary foods commonly appearing as an added sugar [2,3]. This high use in industrially manufactured or confectionary foods can be attributed to the low cost of fructose compared to sucrose. For nearly two decades, the overconsumption of pure fructose has been reported as a potential unique dietary risk factor because of links to obesity risk, the preeminent epidemic and health concern in the US and worldwide [4,5]. The overconsumption of pure fructose is also associated with risk of chronic metabolic noncommunicable diseases such as hypertension [6], cardiovascular diseases [3,7], type 2 diabetes [8], metabolic syndrome [9], steatohepatitis [10] and inflammatory conditions [11]. In this context, overconsumption of pure fructose became an important public health issue in both the scientific and public domain [12,13,14].

Arterial hypertension is one of the top five leading worldwide risks of mortality [15]. It is also the most common modifiable risk factor for cardiovascular disease [16], with the tracking of hypertension across adolescence to adulthood being a very important health metric [17]. Previous studies have suggested that high levels of pure fructose intake contribute to elevation in blood pressure [6,18,19,20,21,22]. This observation has been confirmed in animal studies, where consuming a diet comprising 60% of total energy from fructose induces hypertension [23,24,25,26,27]. Few studies have examined the possible deleterious health effects of pure fructose consumption among adolescents [6], despite the observation that adolescents are the highest consumers of fructose [28,29]. To our knowledge, only two studies have assessed the impact of fructose-rich beverages such as caffeinated and/or sugar-sweetened beverages (SSB) on blood pressure in adolescents [18,30], both showing that consumption of these beverages was associated with higher systolic blood pressure. However, results from these studies were highly criticized because they did not take into account of potential confounding factors such as physical activity level, tobacco consumption and salt intake. Moreover, they did not examine the total all food sources containing fructose [31]; they were only focalized on some food categories such as SSB, for example. The impact of fructose consumption on adolescent blood pressure continues to be controversial [32,33], and the relationship between blood pressure and fructose consumption should be viewed as a major public health issue [34].

The purpose of our study was to assess the association of fructose intake from various foods categories on the blood pressure of adolescents. The hypothesis was that high fructose intake from several food sources extrinsically fructose rich could increase blood pressure in adolescents.

## 2. Materials and Methods

### 2.1. Sample

Data were derived from the Healthy Lifestyle in Europe by Nutrition in Adolescence (HELENA) study from eight different countries in Northern (Ghent in Belgium, Lille in France, Dortmund in Germany, Stockholm in Sweden), Central (Vienna in Austria) and Southern Europe (Athens in Greece, Roma in Italy, Zaragoza in Spain) between 2006 and 2007, as previously described [35]. Briefly, the HELENA study was a multisite study designed to obtain reliable, comparable data about nutritional habits and patterns, body composition and levels of physical activity and fitness from European adolescents aged 12.5 y to 17.5 y. Details of the sampling procedures, field team preparation, the pilot study and data reliability are presented elsewhere [36]. The study was performed in accordance with the ethical guidelines of the Declaration of Helsinki, good clinical practice and the legislation concerning clinical research in each of the participating countries. The protocol was approved by the appropriate independent ethics committee for each study centre, and written informed consent was obtained from both parents and adolescents [37]. In total, 3865 adolescents were enrolled through their schools, which were randomly selected according to a proportional cluster sampling methodology that took age and socioeconomic status into account [38]. From the initial data set of 3528 analysable adolescents, 1705 were complete for fructose intake and blood pressure analysis.

### 2.2. Assessment of Daily Fructose Intake

Dietary intake was assessed by two nonconsecutive 24 h recalls performed at any point in the week [39]. The recalls did not necessarily include a weekday and a weekend day for each individual. The 24 h recalls were recorded with a self-administered, computer-based tool: the HELENA Dietary Intake Assessment Tool (HELENA–DIAT) adapted from the YANA-C tool developed with and validated for Flemish adolescents [40]. The HELENA–DIAT tool is based on six meal occasions (breakfast, morning snacks, lunch, afternoon snacks, evening meal and evening snacks) on the previous day. Trained dieticians assisted the adolescents to complete the 24 h recalls when needed. The adolescents selected all foods and beverages consumed at each meal occasion from a standardized food list [41]. Dietary sources of fructose were extracted according to Mesana et al. [42] and Duffey et al. [43] for SSB. Fructose in its pure monosaccharide form was computed using fructose content data from several studies/databases described in Appendix A [44,45,46,47]. As fructose is also made available through the consumption of sucrose (a disaccharide made of α-D-glucopyranosyl (1→2)-β-D-fructofuranoside), commonly appearing as an added sugar in processed foods [2,3], fructose was also expressed as fructose total exposure. Total fructose exposure was calculated as fructose from monosaccharide form plus fructose from sucrose (0.5 × sucrose per 100 g), in accordance with Ramne et al. [48]. Pure fructose and total fructose exposure were separately analysed because of their difference in intestinal/blood absorption [9]. Two foods categories were created at posteriori using a classification close to Aeberli et al. [49]: (i) food categories extrinsically fructose rich such as industrial/manufactured/confectionary non-natural food products: sugar-sweetened beverages, nonchocolate confectionary, chocolate, cakes/pies/biscuits, desserts and puddings, breakfast cereals and others sources categories [50], and (ii) intrinsically fructose-rich natural foods: fruit/vegetable juices, honey/jam/syrup and whole fruits.

### 2.3. Assessment of Anthropometrics

Weight was measured in underwear, with shoes removed, using an electronic scale (SECA 861, SECA, Birmingham, UK) to the nearest 0.1 kg. Height was measured with shoes removed using a telescopic height measuring instrument (SECA 225) to the nearest 0.1 cm. Body Mass Index (BMI) was calculated by dividing body weight (kg) by the square of height (m^2^), and BMI z-score was calculated using the lambda, mu and sigma method [51].

### 2.4. Assessment of Blood Pressure

Blood pressure (BP) measurements were performed following the recommendations for adolescent population [52]. BP was measured in mmHg twice after weight and height measurements were taken. The subjects were seated in a separate, quiet room for 10 min with their backs supported and feet on the ground. Two BP readings were taken with a 10 min interval of quiet rest. The lower of the two measurements was used. Systolic blood pressure (SBP) and diastolic blood pressure (DBP) were measured by the arm blood pressure oscillometric monitor device OMRON HEALTHCARE HEM7001 (OMRON HEALTHCARE, Koyoto, Japan), which has been approved by the British Hypertension Society [53]. Data collection HEM7001 procedures have been described earlier [36]. The use of percentiles is usually used to define reference clinical data in a set of data from a population. Adolescents of this study were separated in 2 groups according to percentiles of blood pressure data: systolic and diastolic, >90th percentile or ≤90th percentile. Elevated BP was defined where systolic or diastolic blood pressure was >90th percentile of the HELENA population analysed. The cut-off of the 90th percentile to define elevated blood pressure for systolic and diastolic has been described by Flynn, J.T. et al. [54].

### 2.5. Assessment of Physical Activity

Physical activity was measured using a uniaxial accelerometer (ActiGraph GT1M^®^, Pensacola, FL, USA) in pure living conditions [55]. The accelerometer recorded activity for 7 consecutive days the same week as daily fructose intake collection. PA collection by the accelerometer was taken off at night. Moderate to vigorous physical activity (MVPA) time spent was computed when PA counts/min were more than 2000 count/min, cut-off defined by Ekelund et al. [56].

### 2.6. Assessment of Tobacco Consumption

Regular tobacco smoking, defined as the regular consumption of at least one cigarette per day in the past month [57], was assessed using the ad hoc HELENA study questionnaire [36].

### 2.7. Statistical Analysis

Data are presented as frequency (percentage) for categorical variables and mean ± standard deviation (SD) or median interquartile range (IQR) for continuous variables. Normality of distribution was checked graphically and by using the Shapiro–Wilk test. To assess the potential bias related to missing or incomplete nutrients and BP parameters, the main characteristics of included and excluded adolescents were compared using Student’s t-test for continuous variables and Chi-square test for categorical variables. To evaluate the magnitude of differences between analysed and nonanalysed participants, we calculated the absolute standardized differences; a standardized difference >20% denotes a meaningful imbalance. Associations of elevated SBP and DBP values with pure and total fructose exposure (categorized according to tertiles distributions) were investigated with and without adjustment for confounding factors (data of confounding factors are presented in Appendix A). Mixed logistic regression models were used including elevated blood pressure as dependent variables, fructose exposure and the confounding variables as independent fixed effects and centre as a random effect. To avoid case deletion in multivariate analyses, missing data were imputed by multiple imputations using the regression-switching approach (chained equations, m = 10 imputations) [58]. The imputation procedure was performed under the missing-at-random assumption using all variables, with the predictive mean-matching method for continuous variables and logistic regression (binary, ordinal or multinomial) models for categorical variables. Rubin’s rules were used to combine the estimates derived from multiple imputed data sets [58]. Due to established sex differences in prior similar studies of blood pressure in children, analyses were stratified by sex [59,60]. All statistical tests were done at the two-tailed α level of 0.05. Data were analysed with SAS software version 9.4 (SAS Institute Inc., Cary, NC, USA).

## 3. Results

Main characteristics of the 1705 participants are presented in Appendix A. Body mass of excluded adolescents from statistical analysis was slightly higher (ASD = 26.6%) than included adolescents. Consequently, a similar difference (ASD = 35.6%) was found for z-score BMI.

Median pure fructose intake was 34.69 g/day in girls (*n* = 901) and 45.29 g/day in boys (*n* = 804) from all food sources and 12.94 g/day in girls and 21.14 g/day in boys from non-natural foods sources only (Table 1). Total fructose exposure was higher and reached 51.24 g/day in girls and 63.62 g/day in boys from all food sources and 24.47 g/day in girls and 35.89 g/day in boys from non-natural foods sources only. 

Across the two food classifications “all food sources” or “non-natural”, adolescents were classified into low/middle/high tertiles for “pure” and “total fructose” exposure (Table 2).

Table 3 presents mean ± SD of blood pressure in all adolescents and cut-off values for elevated BP or mildly elevated BP. For girls, cut-offs for 90th and 75th percentile were 125 and 118 mmHg for SBP and 75 and 69 mmHg for DBP, respectively. For boys, cut-offs for 90th and 75th percentile were 137 and 127 mmHg for SBP and 75 and 69 mmHg for DBP respectively.

Table 4 shows the association between elevated systolic blood pressure and pure and total fructose exposure from various fructose-containing foods for girls and boys. Elevated blood pressure values were not associated with pure and total fructose exposure from fructose-containing foods.

Table 5 shows the association between elevated diastolic blood pressure and pure and total fructose exposure from various fructose-containing foods. Elevated diastolic blood pressure values were not associated with pure and total fructose exposure from various fructose-containing foods.

Table 6 shows the association between elevated systolic blood pressure and pure and total fructose exposure from non-natural foods. Elevated systolic blood pressure values were not associated with pure and total fructose exposure from various fructose-containing foods.

Table 7 shows the association between elevated diastolic blood pressure and pure and total fructose exposure from non-natural foods for girls and boys. Among the highest tertiles of fructose consumption from non-natural foods, we found an association with elevated diastolic blood pressure in girls (OR = 2.27 (1.17–4.40); *p* = 0.015) that persisted after adjustment for cofounding factors compared to the lowest tertiles of “pure” fructose.

## 4. Discussion

Fructose intake has quadrupled in the US since the beginning of the 20th century [61,62] and is probably rising across Europe [42], a trend which is predominately driven by the consumption of processed and non-natural food products such as SSB [13]. The median pure and total fructose exposures from our study were similar to the 2007–2010 Dutch food consumption survey (46 g/day) in the same age group [46] and an earlier NHANES study (59 g/day) [63], but greater than a similar study in New Zealander adolescents (21.6 g/day for boys and 18.3 g/day for girls) [46]. In these prior studies, the higher fructose intake observed was explained by SSB consumption [46,63]. Other high-fructose-containing foods contributing to total fructose exposure were honey/jam/syrup, fruits, fruit/vegetable juices and cakes/pies/biscuits, which was also observed in a study of adolescents from Switzerland [49]. In this context, fructose from SSBs could be considered the main source of pure fructose exposure in the typical adolescent diet [62]. Therefore, we consider the data we obtained in the HELENA study to represent habitual pure fructose intake of European adolescent. Most studies showed that medians of fructose intake in subjects more than 15 years old was 50 g/day [9,64,65]. The cut-off of 50 g/day is close to the cut-off of excessive fructose intake used in our study: 41.90 g/day for girls and 54 g/day for boys. The use of tertiles to find the cut-off of excessive fructose (the third tertile) intake in our specific adolescent population permits a well-balanced sample size between groups according to a more powerful statistical analysis.

The effect of fructose intake in the present study was analysed accounting for cofounding factors known to mediate blood pressure [60,66], as demonstrated in similar studies [18,66,67]. Other influencing factors have been analysed such as menarche, presence or absence of menstruation cycle (within one week before BP measurement) and contraceptive use. Data did not show any influence of menarche, presence or absence of menstruation cycle or contraceptive use on girls diastolic BP (using Chi-square tests, data not shown).

In our study, cut-off of 90th percentiles for elevated SBP and DBP were close to the study described by Flynn, J.T. et al. [54]. Adolescence is considered as a vulnerable period of high blood pressure development because the highest peak of blood pressure occurs during puberty [68,69]. As increase of adrenal androgen production occurs earlier in girls than in boys, pure fructose intake levels have more impact on DBP and were significant in girls from this study population [70].

DBP differed among pure fructose intake levels (from non-natural foods) in adolescent girls but not in boys. Indeed, during girls’ puberty, there is an increase of adrenal androgen production [71]. Additionally, girls are less physically active than boys at this age [72], which could contribute to elevated DBP. Moreover, presence of polycystic ovary syndrome, which is now frequently associated with an increase of DBP in adolescent girls, could be a possible explanation.

The proposed mechanism by which fructose intake increases blood pressure after an acute load is related to fructose metabolism. After an acute load, maximum blood fructose levels are rapidly achieved at 60 min [65]. Pure fructose is degraded in the liver, resulting in a rapid and transient increase of blood uric acid levels. Importantly, fructose is the only carbohydrate that increases blood uric acid in humans, and uric acid exerts hemodynamic effects (increased oxidative stress, endothelial dysfunction and activation of the renin–angiotensin–aldosterone system) that have been shown to contribute to high blood pressure [6,20,21,23]. Indeed, there are epidemiologic/clinical data and plausible mechanisms that explain increased blood pressure with excessive fructose intake (more than 40 or 50 g/day) [73]. In the high-fructose, non-natural-foods-consumers groups from HELENA study, we observed elevated diastolic blood pressure in adolescent girls. This may be explained by fructose from fruits and vegetables being more slowly absorbed, compared to pure fructose, due to the presence of dietary fibres that slow fructose metabolism [74].

There are several strengths in the present study. Most studies on fructose consumption have focused on SSB intake [18,28,75,76,77], whereas our study takes into account the contribution of several other food types to fructose exposure. The HELENA study is comprised of a large sample size from eight geographically diverse European cities, and fructose exposure was collected in a “real life” manner from validated dietary instruments. Dietary intake was collected, across all eight countries, using these standardized and validated tools (HELENA–DIAT) and the same food composition database, allowing us to assess both between- and within-individual variability. The large battery of health data, lifestyle measures, anthropometric data and SES variables measured in the HELENA adolescents were obtained through standardized and validated procedures/tools, allowing for analyses to be adjusted for several possible confounding factors. There are some limitations to the present study. Due to the cross-sectional design of the study, causality cannot be determined. Pure fructose consumption was collected from dietary records representing 2 days; thus, it may not completely reflect habitual adolescent consumption patterns and allow total quantification of fructose exposure. Similarly, the fructose content of foods comes from multiple food data composition database or datasets, as a comprehensive dataset on the content of fructose in foods does not exist. Unfortunately, the question about the presence of polycystic ovary syndrome could not been answered. Indeed, confirmation of this diagnostic should include a hormone blood sample analysis and an ovarian ultrasound imaging, which were not performed in the HELENA study. Differences in intake of fructose in the various countries were not assessed in the HELENA study. Some blood sample analyses such as renin or aldosterone were not performed, and only one-third of adolescents’ Hb1Ac was analysed [38]. Lastly, our study did not record history of parental hypertension and genetic factors associated with BP; therefore, we could not use these data in our analyses.

## 5. Conclusions

In conclusion, consuming high quantities of non-natural foods containing extrinsically high fructose is associated with elevated DBP in adolescent girls. The consumption of natural foods containing fructose, such as whole fruits, does not impact blood pressure in our study and should continue to remain a healthy dietary habit [78,79,80], especially since whole fruit could decrease diastolic blood pressure in girls [81].

## Figures and Tables

**Table 1 nutrients-13-03608-t001:** Daily fructose intake from various fructose-containing foods in girls and boys.

**Girls**	**Foods**	**Pure Fructose** **(g/day)**	**Total Fructose** **(g/day)**
	Sugar-sweetened beverages	7.98 (2.51 to 17.22)	8.80 (2.76 to 18.98)
	Nonchocolate confectionary	0.12 (0.05 to 0.28)	1.00 (0.40 to 2.41)
	Chocolate	1.53 (0.67 to 2.73)	1.71 (0.75 to 3.05)
	Cakes/pies/biscuits	1.95 (1.07 to 2.90)	8.79 (4.83 to 13.10)
	Desserts and puddings	0.09 (0.06 to 0.19)	0.22 (0.14 to 0.47)
	Breakfast and cereals	0.01 (0.01 to 0.20)	0.13 (0.07 to 1.90)
	Others	0.02 (0.01 to 0.02)	0.02 (0.01 to 0.02)
	Fruit and vegetable juices	5.14 (1.85 to 10.08)	5.86 (2.10 to 11.49)
	Honey/jam/syrup	1.31 (0.37 to 5.28)	1.61 (0.45 to 6.50)
	Fruits	8.60 (4.62 to 13.70)	10.45 (5.61 to 16.64)
	Fructose from all food sources	34.69 ** (25.39 to 46.35)	51.24 ** (39.17 to 65.12)
	Fructose from non-natural foods *	12.94 ** (7.57 to 22.36)	25.47 ** (18.04 to 36.76)
**Boys**	**Foods**	**Pure Fructose** **(g/day)**	**Total Fructose** **(g/day)**
	Sugar-sweetened beverages	15.10 (6.04 to 27.45)	16.64 (6.66 to 30.25)
	Nonchocolate confectionary	0.06 (0.03 to 0.20)	0.55 (0.28 to 1.78)
	Chocolate	1.71 (0.91 to 3.74)	1.91 (1.01 to 4.18)
	Cakes/pies/biscuits	2.12 (1.03 to 3.41)	9.58 (4.64 to 15.40)
	Desserts and puddings	0.05 (0.04 to 0.08)	0.12 (0.09 to 0.19)
	Breakfast and cereals	0.03 (0.02 to 0.28)	0.25 (0.15 to 2.70)
	Others	0.01 (0.01 to 0.02)	0.01 (0.01 to 0.02)
	Fruit and vegetable juices	5.43 (1.94 to 11.19)	6.19 (2.21 to 12.75)
	Honey/jam/syrup	1.40 (0.61 to 5.81)	1.73 (0.76 to 7.15)
	Fruits	8.05 (3.59 to 13.29)	9.78 (4.37 to 16.13)
	Fructose from all food sources	45.29 ** (32.19 to 59.98)	63.62 ** (47.55 to 81.12)
	Fructose from non-natural foods *	21.14 ** (11.69 to 34.60)	35.89 ** (23.61 to 51.55)

Values are medians and interquartile range * Non-natural foods are: sugar-sweetened beverages, nonchocolate confectionary, chocolate, cakes/pies/biscuits, desserts and puddings, breakfast cereals and others sources categories. ** values presented are medians; the sum of medians from each foods categories do not follow an arithmetic computation.

**Table 2 nutrients-13-03608-t002:** Tertiles of daily pure fructose intake from all food sources and from non-natural foods sources.

**Tertiles Levels** **For Girls**	** *n* ** **(%)**	**Ranges for Pure Fructose** **(g/day)**	***n* (%)**	**Ranges for Total Fructose Exposure (g/day)**
All food sources				
Low	284 (31.5%)	7.85 to 27.60	285 (31.6%)	13.78 to 41.88
Middle	311 (34.5%)	27.63 to 41.76	310 (34.4%)	41.92 to 59.38
High	306 (34.0%)	41.90 to 130.86	306 (34.0%)	59.52 to 160.16
Non-natural foods				
Low	296 (32.9%)	1.39 to 9.12	296 (32.8%)	3.80 to 20.18
Middle	305 (33.8%)	9.15 to 18.73	304 (33.7%)	20.22 to 32.01
High	300 (33.3%)	18.76 to 81.50	301 (33.4%)	32.15 to 112.56
**Tertiles Levels** **For Boys**	***n* (%)**	**Ranges for Pure Fructose** **(g/day)**	***n* (%)**	**Ranges for Total Fructose Exposure (g/day)**
All food sources				
Low	258 (32.1%)	9.55 to 35.65	256 (31.8%)	13.17 to 51.76
Middle	270 (33.6%)	35.65 to 53.98	273 (34.0%)	51.77 to 72.70
High	276 (34.3%)	54.00 to 136.75	275 (34.2%)	72.83 to 191.25
Non-natural foods				
Low	267 (33.2%)	2.21 to 14.15	269 (33.5%)	4.56 to 27.55
Middle	272 (33.8%)	14.19 to 28.92	267 (33.2%)	27.61 to 45.01
High	265 (33.0%)	28.9 to 114.79	268 (33.3%)	45.07 to 161.18

**Table 3 nutrients-13-03608-t003:** Blood pressure data and threshold for elevated blood pressure.

**Blood Pressure for Girls**	** *n* **	**Mean ± SD** **(mmHg)**
Systolic blood pressure (SBP)	901	111.56 ± 11.21
Elevated SBP above the 90th percentile > 125 (mmHg)	84	132.52 ± 7.27
Mildly elevated above the 75th percentile > 118 (mmHg)	223	125.7 ± 7.1
Mildly elevated above 110 (mmHg)	487	119.6 ± 7.6
Diastolic blood pressure (DBP)	901	64.48 ± 8.50
Elevated DBP above the 90th percentile > 75 (mmHg)	86	80.52 ± 6.10
Mildly elevated above the 75th percentile > 69 (mmHg)	228	75.4 ± 5.7
Mildly elevated above 70 (mmHg)	199	76.1 ± 5.7
**Blood Pressure for Boys**	** *n* **	**Mean ± SD** **(mmHg)**
Systolic blood pressure (SBP)	804	119.61 ± 13.40
Elevated SBP above the 90th percentile > 137 (mmHg)	79	145.32 ± 7.85
Mildly elevated above the 75th percentile > 127 (mmHg)	203	137.1 ± 8.5
Mildly elevated above 110 (mmHg)	600	125.1 ± 10.6
Diastolic blood pressure (DBP)	804	63.99 ± 8.41
Elevated DBP above the 90th percentile > 75 (mmHg)	86	79.21 ± 4.33
Mildly elevated above the 75th percentile > 69 (mmHg)	197	75.0 ± 4.8
Mildly elevated above 70 (mmHg)	170	75.8 ± 4.7

Abbreviations: SBP = systolic blood pressure; DBP = diastolic blood pressure.

**Table 4 nutrients-13-03608-t004:** Association between elevated systolic blood pressure and pure and total fructose exposure from various fructose-containing foods.

		**Model 1**		**Model 2**	
**For Girls**	**Elevated SBP** **(*n* = 84)**	**OR (95%CI)**	** *p* **	**OR (95%CI)**	** *p* **
Pure fructose			0.97 *		0.66 *
Low	22/284 (7.7%)	1.00 (ref.)	-	1.00 (ref.)	-
Middle	22/311 (7.1%)	0.72 (0.37 to 1.39)	0.32	0.76 (0.38 to 1.53)	0.45
High	40/306 (13.1%)	0.96 (0.51 to 1.84)	0.91	1.14 (0.55 to 2.39)	0.72
Total fructose exposure			0.48 *		0.86 *
Low	24/285 (8.4%)	1.00 (ref.)	-	1.00 (ref.)	-
Middle	21/310 (6.8%)	0.67 (0.35 to 1.28)	0.23	0.80 (0.40 to 1.60)	0.54
High	39/306 (12.7%)	0.83 (0.44 to 1.57)	0.57	1.06 (0.50 to 2.21)	0.89
		**Model 1**		**Model 2**	
**For Boys**	**Elevated SBP** **(*n* = 79)**	**OR (95%CI)**	** *p* **	**OR (95%CI)**	** *p* **
Pure fructose			0.12 *		0.035 *
Low	24/258 (9.3%)	1.00 (ref.)	-	1.00 (ref.)	-
Middle	25/270 (9.3%)	0.72 (0.38 to 1.34)	0.30	0.55 (0.27 to 1.09)	0.084
High	30/276 (10.9%)	0.60 (0.31 to 1.14)	0.12	0.44 (0.21 to 0.93)	0.031
Total fructose exposure			0.44 *		0.13 *
Low	21/256 (8.2%)	1.00 (ref.)	-	1.00 (ref.)	-
Middle	28/273 (10.3%)	0.96 (0.51 to 1.80)	0.89	0.64 (0.32 to 1.30)	0.22
High	30/275 (10.9%)	0.78 (0.41 to 1.50)	0.46	0.54 (0.25 to 1.17)	0.12

High SBP is defined as a value greater than the 90th percentile (>125 mmHg for girls and >137 mmHg for boys). Model 1: adjusted for centre. Model 2: adjusted for centre, age, Z-score BMI, MVPA duration, tobacco consumption, salt intake and energy intake and calculated after multiple imputations (m = 10) to handle missing data. * *p* calculated using fructose levels as ordinal variable. Abbreviations: SBP = systolic blood pressure; OR = odds-ratio; CI = confidence interval.

**Table 5 nutrients-13-03608-t005:** Association between elevated diastolic blood pressure and pure and total fructose exposure from various fructose-containing foods.

		**Model 1**		**Model 2**	
**For Girls**	**Elevated DBP** **(*n* = 86)**	**OR (95%CI)**	** *p* **	**OR (95%CI)**	** *p* **
Pure fructose			0.12 *		0.038 *
Low	23/284 (8.1%)	1.00 (ref.)	-	1.00 (ref.)	-
Middle	18/311 (5.8%)	0.68 (0.35 to 1.33)	0.26	0.67 (0.34 to 1.34)	0.26
High	45/306 (14.7%)	1.51 (0.81 to 2.78)	0.19	1.93 (0.99 to 3.75)	0.052
Total fructose exposure			0.45 *		0.086 *
Low	23/285 (8.1%)	1.00 (ref.)	-	1.00 (ref.)	-
Middle	23/310 (7.4%)	0.89 (0.47 to 1.66)	0.71	1.10 (0.57 to 2.10)	0.78
High	40/306 (13.1%)	1.24 (0.67 to 2.31)	0.50	1.83 (0.91 to 3.65)	0.089
**For Boys**		**Model 1**		**Model 2**	
	**Elevated DBP** **(*n* = 86)**	**OR (95%CI)**	** *p* **	**OR (95%CI)**	** *p* **
Pure fructose			0.88 *		0.38 *
Low	25/258 (9.7%)	1.00 (ref.)	-	1.00 (ref.)	-
Middle	29/270 (10.7%)	1.02 (0.57 to 1.83)	0.95	0.87 (0.45 to 1.68)	0.69
High	32/276 (11.6%)	0.96 (0.52 to 1.77)	0.89	0.72 (0.34 to 1.51)	0.39
Total fructose exposure			0.45 *		0.97 *
Low	22/256 (8.6%)	1.00 (ref.)	-	1.00 (ref.)	-
Middle	29/273 (10.6%)	1.11 (0.61 to 2.03)	0.73	0.98 (0.49 to 1.94)	0.95
High	35/275 (12.7%)	1.26 (0.68 to 2.34)	0.46	1.01 (0.47 to 2.16)	0.98

High DBP is defined as a value upper than the 90th percentile (>75 mmHg). Model 1: adjusted for centre. Model 2: adjusted for centre, age, Z-score BMI, MVPA duration, tobacco consumption, salt intake and energy intake and calculated after multiple imputations (m = 10) to handle missing data. * *p* calculated using fructose levels as ordinal variable. Abbreviations: DBP = diastolic blood pressure; OR = odds-ratio; CI = confidence interval.

**Table 6 nutrients-13-03608-t006:** Association between elevated systolic blood pressure with pure and total fructose exposure from non-natural foods.

		**Model 1**		**Model 2**	
**For Girls**	**Elevated SBP** **(*n* = 84)**	**OR (95%CI)**	** *p* **	**OR (95%CI)**	** *p* **
Pure fructose			0.79 *		0.70 *
Low	21/296 (7.1%)	1.00 (ref.)	-	1.00 (ref.)	-
Middle	29/305 (9.5%)	1.01 (0.54 to 1.88)	0.98	1.09 (0.56 to 2.10)	0.80
High	34/300 (11.3%)	0.92 (0.49 to 1.76)	0.81	1.15 (0.57 to 2.32)	0.70
Total fructose exposure			0.22 *		0.38 *
Low	26/296 (8.8%)	1.00 (ref.)	-	1.00 (ref.)	-
Middle	26/304 (8.6%)	0.79 (0.43 to 1.44)	0.44	1.03 (0.52 to 2.04)	0.93
High	32/301 (10.6%)	0.67 (0.36 to 1.26)	0.22	1.45 (0.69 to 3.04)	0.33
**For Boys**		**Model 1**		**Model 2**	
	**Elevated SBP** **(*n* = 79)**	**OR (95%CI)**	** *p* **	**OR (95%CI)**	** *p* **
Pure fructose			0.75 *		0.52 *
Low	19/267 (7.1%)	1.00 (ref.)	-	1.00 (ref.)	-
Middle	30/272 (11.0%)	1.15 (0.61 to 2.18)	0.66	1.04 (0.53 to 2.06)	0.90
High	30/265 (11.3%)	0.93 (0.48 to 1.81)	0.83	0.81 (0.39 to 1.67)	0.56
Total fructose exposure			0.83 *		0.45 *
Low	21/269 (7.8%)	1.00 (ref.)	-	1.00 (ref.)	-
Middle	28/267 (10.5%)	1.02 (0.54 to 1.93)	0.94	0.68 (0.35 to 1.33)	0.26
High	30/268 (11.2%)	0.94 (0.49 to 1.79)	0.85	0.83 (0.39 to 1.76)	0.62

High SBP is defined as a value greater than the 90th percentile (>125 mmHg for girls and >137 mmHg for boys)). Model 1: adjusted for centre. Model 2: adjusted for centre, age, Z-score BMI, MVPA duration, tobacco consumption, salt intake and energy intake and calculated after multiple imputations (m = 10) to handle missing data. * *p* calculated using fructose levels as ordinal variable. Abbreviations: SBP = systolic blood pressure; OR = odds-ratio; CI = confidence interval.

**Table 7 nutrients-13-03608-t007:** Association between elevated diastolic blood pressure with pure and total fructose exposure from non-natural foods.

		**Model 1**		**Model 2**	
**For Girls**	**Elevated DBP** **(*n* = 86)**	**OR (95%CI)**	** *p* **	**OR (95%CI)**	** *p* **
Pure fructose			0.051 *		0.013 *
Low	19/296 (6.4%)	1.00 (ref.)	-	1.00 (ref.)	-
Middle	26/305 (8.5%)	1.23 (0.65 to 2.33)	0.52	1.36 (0.71 to 2.59)	0.36
High	41/300 (13.7%)	1.82 (0.97 to 3.39)	0.061	2.27 (1.17 to 4.40)	0.015
Total fructose exposure			0.18 *		0.030 *
Low	22/296 (7.4%)	1.00 (ref.)	-	1.00 (ref.)	-
Middle	24/304 (7.9%)	0.97 (0.52 to 1.80)	0.92	2.06 (1.14 to 3.70)	0.016
High	40/301 (13.3%)	1.47 (0.81 to 2.66)	0.21	1.85 (0.91 to 3.79)	0.091
		**Model 1**		**Model 2**	
**For Boys**	**Elevated DBP** **(*n* = 86)**	**OR (95%CI)**	** *p* **	**OR (95%CI)**	** *p* **
Pure fructose			0.73 *		0.87 *
Low	24/267 (9.0%)	1.00 (ref.)	-	1.00 (ref.)	-
Middle	30/272 (11.0%)	1.06 (0.59 to 1.91)	0.84	1.04 (0.53 to 2.01)	0.91
High	32/265 (12.1%)	1.11 (0.60 to 2.06)	0.73	1.06 (0.52 to 2.16)	0.86
Total fructose exposure			0.71 *		0.86 *
Low	25/269 (9.3%)	1.00 (ref.)	-	1.00 (ref.)	-
Middle	29/267 (10.9%)	1.02 (0.57 to 1.84)	0.94	0.82 (0.42 to 1.59)	0.55
High	32/268 (11.9%)	1.12 (0.61 to 2.04)	0.72	1.17 (0.55 to 2.49)	0.68

High DBP is defined as a value greater than the 90th percentile (>75 mmHg). Model 1: adjusted for centre. Model 2: adjusted for centre, age, Z-score BMI, MVPA duration, tobacco consumption, salt intake and energy intake and calculated after multiple imputations (m = 10) to handle missing data. * *p* calculated using fructose levels as ordinal variable. Abbreviations: DBP = diastolic blood pressure; OR = odds-ratio; CI = confidence interval.

## Data Availability

Not applicable.

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
