# Peer review of "High Fructose Intake Contributes to Elevated Diastolic Blood Pressure in Adolescent Girls: Results from The HELENA Study"

_nutrients, 2021, doi:10.3390/nu13103608_

Round 1

Reviewer 1 Report

Previous studies have suggested that high levels of pure fructose intake contribute to elevation in blood pressure. The study has evaluated the possible relationship between fructose intake and blood pressure in adolescent male and female. The adolescents selected all foods and beverages consumed at each meal occasion from a standardized food list. Dietary sources of fructose were extracted according to Mesana et al and Duffey et al for SSB, fructose was also expressed as fructose total exposure. Consuming high quantities of non-natural foods was associated with elevated DBP in adolescent girls, which was in part due to high fructose levels in these foods categories while the consumption of natural foods containing fructose, such as whole fruits, does not impact blood pressure and should continue to remain a healthy dietary habit

comments

-please report any presence or absence of menarche or of contraceptive use (which increases diastolic BP).

-Was previous physical activity investigated or was physical activity only evaluated in one week?. Was the physical activity done in the same days of the evaluation of fructose intake?

-Are there differences in intake of fructose in the various countries involved in the study?

-What is the criterion for deciding the cut offs for high average and low fructose intake? A previous meta-analysis had considered that undesired effects occurred with an intake higher than 100 grams per day and that a value between 0 and 80 grams did not give health risks.

-Animal studies have shown that high-fructose diets up-regulate sodium and chloride transporters, resulting in a state of salt overload that increases blood pressure. Excess fructose has also been found to activate vasoconstrictors, inactivate vasodilators, and over-stimulate the sympathetic nervous system. There are studies or renin and aldosterone measurement under high fructose diet?

-The results section should be expanded reporting the results of the various tables

-Explain the possible reason for the different diastolic pressure response in males and females

-The authors report as a possible explanation the increase in androgens typical of puberty in females and that this may be associated with a more pronounced fructose effect of the diastolic. This point needs to be developed as 15% of women at this age have polycystic ovary syndrome which is associated with insulin resistance and hyperandrogenism. How many had PCOS?  It is known that PCOS is frequently associated with increased diastolic and that usually patients with PCOS have greater tendency for physical and competitive activity. This could explain the association between prevalence in girls and not in boys where the androgenic trigger occurs later.     It is also worth of note that often aldosterone or the aldosterone renin ratio is increased in PCOS, and this finding could contribute to the increase in diastolic regardless of the amount of fructose ingested.

-The part that discusses cardiovascular risk should be toned down since the problem usually occurs later and involves both genetic and environmental factors.

Reviewer 2 Report

Authors use only 10% of their study population to test if fructose intake is associated with Blood Pressure.  I would ask that these authors consider using a lower threshold of the population as a normal BP in youth is likely 100/60.  Please consider using the 60th or 70th percentile for their analysis.   

An additional evaluation may be by using the 100/70 threshold to mildly elevated BP in youth or 110/70. 

Reviewer 3 Report

The impact of fructose consumption on adolescent blood pressure remains to be controversial. This manuscript reports results derived from the Healthy Lifestyle in Europe by Nutrition in Adolescence (HELENA) study in eight European countries to suggest that the consumption of high quantities of non-natural foods was associated with elevated diastolic blood pressure in adolescent girls, which was in part due to high fructose levels in these foods categories. This is a reasonably well executed study reporting data that support the conclusion. Comments and suggestions are provided below.       

General comments:

Two major concerns regarding experimental design: (1) no biochemical data on metabolic parameters (e.g., fasting blood glucose, plasma lipid profile, Hb-A1c …) of the subjects; (2) no information on assessment of daily salt intake. Both points may impact blood pressure values of the study. These limitations need to be discussed.

Specific comments:

  1. Line 74, define SSB when appear firstly in the text.
  2. Lines 149-150, please be more specific on grouping of the subjects. What specific percentile of blood pressure (SBP or DBP?) was used to separate the subjects into 9 groups? It is also unclear the significance of this grouping on data presented in the Results section.
  3. Line 166, change blood pressure to BP. Double check to confirm that all abbreviations are properly used in the text.
  4. Lines 302-303, please define SSB on Line 74 and use the abbreviation here.
  5. Lines 319-320, were the measurements of blood pressure taken during or between menstrual cycles in female adolescents? Would the BP readings be different if taken during the menstrual period?

Round 2

Reviewer 1 Report

the Authors have answered to tyhe questioons

Reviewer 2 Report

Accept with changes make.